# Impact of Dextran-Sodium-Sulfate-Induced Enteritis on Murine Cytomegalovirus Reactivation

**DOI:** 10.3390/v14122595

**Published:** 2022-11-22

**Authors:** Alexandre Jentzer, Sébastien Fauteux-Daniel, Paul Verhoeven, Aymeric Cantais, Melyssa Yaugel Novoa, Fabienne Jospin, Blandine Chanut, Nicolas Rochereau, Thomas Bourlet, Xavier Roblin, Bruno Pozzetto, Sylvie Pillet

**Affiliations:** 1CIRI, Centre International de Recherche en Infectiologie, GIMAP Team, Univ Lyon, Univ St-Etienne, INSERM U1111, CNRS UMR5308, ENS de Lyon, UCBL1, 42023 Saint-Etienne, France; 2French Blood Establishment Auvergne-Rhône-Alpes, Scientific Department, 42270 Saint-Etienne, France; 3Department of Infectious Agents and Hygiene, University-Hospital of Saint-Etienne, 42055 Saint-Etienne, France; 4Department of Gastroenterology, University-Hospital of Saint-Etienne, 42055 Saint-Etienne, France

**Keywords:** murine cytomegalovirus, mouse model, viral reactivation, DSS-induced enteritis

## Abstract

(1) Background: Ulcerative colitis (UC) is an inflammatory bowel disease that causes inflammation of the intestines, which participates in human cytomegalovirus (HCMV) reactivation from its latent reservoir. CMV-associated colitis plays a pejorative role in the clinical course of UC. We took advantage of a model of chemically induced enteritis to study the viral reactivation of murine CMV (MCMV) in the context of gut inflammation. (2) Methods: Seven-week-old BALB/c mice were infected by 3 × 10^3^ plaque-forming units (PFU) of MCMV; 2.5% (*w*/*v*) DSS was administered in the drinking water from day (D) 30 to D37 post-infection to induce enteritis. (3) Results: MCMV DNA levels in the circulation decreased from D21 after infection until resolution of the acute infection. DSS administration resulted in weight loss, high disease activity index, elevated Nancy index shortening of the colon length and increase in fecal lipocalin. However, chemically induced enteritis had no impact on MCMV reactivation as determined by qPCR and immunohistochemistry of intestinal tissues. (4) Conclusions: Despite the persistence of MCMV in the digestive tissues after the acute phase of infection, the gut inflammation induced by DSS did not induce MCMV reactivation in intestinal tissues, thus failing to recapitulate inflammation-driven HCMV reactivation in human UC.

## 1. Introduction

Inflammatory bowel diseases (IBD) include Crohn’s disease and ulcerative colitis (UC), both associated with inflammation of the digestive tract [1]. UC incidence and prevalence is increasing worldwide, which results in a major public health issue [2,3,4]. UC is clinically characterized by the succession of acute inflammatory flares and remission phases with major consequences on the socio-professional life of patients [5]. Despite UC incurability, the administration of anti-inflammatory and/or immunosuppressive therapies allows, in most cases, the control of acute phases and broadening of remission periods [6].

Human cytomegalovirus (HCMV) is an opportunistic pathogen of the *Betaherpesvirinae* subfamily. A meta-analysis published in 2019 estimated the global seroprevalence of HCMV in adults at 83% [95% CI (confidence interval): 78–88] [7]. HCMV primary infection is characterized by viremia followed by systemic dissemination. HCMV can replicate in many cell types, including endothelial cells, epithelial cells, fibroblasts, and monocytes/macrophages [8]. After control of the acute phase, HCMV enters into a latency state in hematopoietic progenitors and other cells disseminated in various tissues [9] that serve as viral reservoirs from which new viral particles are produced following viral reactivation, notably in the course of an inflammation process. Although HCMV infection is most often contained by the immune system, disease can occur with severe impairment on the functions of targeted organs including brain, lungs and digestive tract, especially in immunocompromised patients [8]. Viral reactivation also occurs in patients suffering from UC, which mainly results in HCMV-associated colitis [10]. HCMV gut reactivation aggravates the course of UC [11] with an increased risk of colectomy [12,13,14] and mortality [12,15] for these patients. We and others have documented the deleterious effect of HCMV reactivation in the context of corticoids and immunosuppressive therapies in the course of UC flare-ups [16,17,18]. To date, the pathophysiology of viral reactivation is poorly understood in UC patients, and animal models are lacking to contribute to this characterization [19,20].

The aim of the present study was to establish a mouse model of murine cytomegalovirus (MCMV) reactivation in the context of chemically induced enteritis mimicking HCMV reactivation in human colitis flare-ups. BALB/c mice were first infected by MCMV. After resolution of the acute phase, gut inflammation was induced by the oral administration of dextran sodium sulfate (DSS) that acts as a trigger for acute enteritis [21].

## 2. Materials and Methods

### 2.1. Mice, Viral Strain and Stock

BALB/c mice were hosted at the PLEXAN (Platform for Experiments and Analysis, Faculty of Medicine, University of Saint-Etienne, France) which is a conventional animal facility with infectious sector P2. MCMV Smith strain (ATCC-VR-1399) was amplified by infecting 7-week-old mice with 1 × 10^4^ plaque-forming units (PFU) administered by the intraperitoneal (IP) route. Salivary glands were collected 3 weeks post-infection (p.i.), and the viral titration was performed on M2-10B4 cells (ATCC^®^ CRL-1972™) [22]; viral titre was determined as PFU/mL. The viral stock was aliquoted and stored frozen at −80 °C until use.

### 2.2. Experimental Design

Four groups of mice (*n* = 5, 7-week-old females per group) were distributed as follows: negative controls (group 1), mice undergoing dextran-sodium-sulfate (DSS) treatment (group 2), MCMV-infected mice (group 3), and MCMV-infected mice challenged by DSS treatment (group 4). At day 0 (D0), groups 3 and 4 were infected (IP) with 200 µL of a suspension of MCMV containing 3 × 10^3^ PFU; groups 1 and 2 were injected (IP) with 200 μL of phosphate buffered saline (PBS). DSS (36–50 KDa, MP Biomedicals™, Irvine, CA, USA) was diluted at 2.5% (*w*/*v*) in PBS and filtered < 0.22 μm before administration; DSS was administered in the drinking water of groups 2 and 4 from day (D) 30 to D37 and renewed daily from D30 to D37 as previously described [19]. The experimental design is summarized in Figure 1. Blood samples were collected by the submandibular route at D0, D7, D14, D21, D30 and D37 p.i. On D37, the mice were euthanized; blood and samples of the salivary glands, small intestine and colon were collected. To assess the macroscopic impact of DSS administration on the colon, its length was measured from the cecum to the rectum [23].

### 2.3. Clinical Score

Each animal was examined daily from D30 to D37 and evaluated by a validated score called the Disease Activity Index (DAI) [23], including weight loss scored from 0 to 4 from baseline (0 < 2%, 1 = [2–5%], 2 = [5–10%], 3 = [10–20%], and 4 > 20%), stool consistency scored from 0 to 4 (0 = normal stools; 2 = loose stools and 4 = acute diarrhoea), and the presence of fecal blood scored from 0 to 2 (0 = no blood, 1 = moderate blood, and 2 = gross bleeding and/or blood clots), on total scale of 10 points

### 2.4. Measure of Fecal Lipocalin

Mouse stools were collected, weighed and stored with a protease inhibitor (Halt Protease Inhibitor Cocktail, Thermo Fisher Scientific, Waltham, MA, USA). They were homogenized for 1 h and centrifuged at 10,000× *g* for 10 min. The supernatant was decanted and stored at −20 °C until analysis. The fecal lipocalin, recognized to assess intestinal inflammation by non-invasive manner [24], was measured by enzyme-linked immunosorbent assay (ELISA) using the Mouse Lipocalin-2/NGAL DuoSet ELISA kit (Bio-Techne, Minneapolis, MN, USA) according to the supplier’s recommendations. 

### 2.5. Collection of Organ Samples

On the day of euthanasia (D37), organs mentioned above were dissected and immersed in PBS. The endoluminal part of the small intestine and colon were washed with PBS. The tissues were fixed in 4% paraformaldehyde (PFA), including an injection into the endoluminal part of the small intestine and colon. Specimens of salivary glands, small intestine and colon were included in optimal cutting temperature (OCT) (Sigma Aldrich, St. Louis, MO, USA) and stored at −20 °C. Eight micrometer sections were made with a cryostat (Leica CM1950) and mounted on SuperFrost™ slides (Thermo Fisher Scientific, Waltham, MA, USA).

### 2.6. Histopathology 

The hematoxylin and eosin (H&E) stained sections of the small intestine and colon were used to measure the histological activity of DSS-induced enteritis using the Nancy index as previously described for humans and mice [25,26]: grade 0 = healthy mucosa, grade 1 = moderate to severe, increased number of infiltrating cells and the absence of inflammatory elements, grade 2 = mucosa with rare neutrophils in the lamina propria and/or in the epithelium and the absence of ulceration, grade 3 = mucosa with moderate to severe infiltration by neutrophils in the lamina propria and/or in the epithelium and the absence of ulceration, grade 4 = presence of ulceration.

### 2.7. Immunohistochemistry

Active replication of MCMV was detected as follows. Active replication of MCMV was detected in the colon and small intestine by staining Immediate-early-1 MCMV protein (IE1) using Primary monoclonal mouse anti-m123/ IE1 antibody (IgG2a kappa, CAPRI HR-MCMV-12, Rijeka, Croatia). Anti-m123/IE1 was diluted 1:500 in PBS and incubated on sections for 1 h at 37 °C f. Primary monoclonal antibody against alpha smooth muscle actin (anti-αSMA) (IgG2a kappa, Abcam, Cambridge, UK) was diluted 1:200 in PBS and incubated for 1 h at 37 °C for actin labelling as positive control. The primary antibodies were biotinylated using Amicon^®^ kit (Millipore, Burlington, MA, USA). After washing steps, staining was achieved by adding horseradish peroxidase-conjugated streptavidin (Pharmingen™, Becton Dickinson, Franklin Lakes, NJ, USA) for 30 min at room temperature away from light. Sections were further washed, and chromogenic diaminobenzidine substrate (Pharmingen™) was added for 5 min at room temperature away from light. The immunostaining protocol was validated using the salivary glands of BALB/c mice infected with 3 × 10^3^ PFU of MCMV by the IP route and dissected on D21 (Appendix A); the negative control of the experiment (performed without primary antibody) showed no unspecific labeling.

### 2.8. MCMV qPCR in Mouse Blood and Tissues

Blood was collected in EDTA coated tubes, and Halt Protease Inhibitor Cocktail (Thermo Fisher Scientific) was added to the blood samples. MCMV DNA extraction was performed using NUCLISENS^TM^ EasyMAG^®^ (bioMérieux, Craponne, France) as previously described for human whole blood [27]. MCMV qPCR assay was performed according to the supplier’s recommendations (Applied Biosystems 7500, Thermo Fisher Scientific). A fragment of the gene encoding MCMV glycoprotein B (gB) was amplified by using the Platinum^®^ SYBR^®^Green Quantitative PCR SuperMix-UDG (Invitrogen^TM^, Thermo Fisher Scientific) and the sequence of primers were the following: 5′-AGG-CCG-GTC-GAG-TAC-TTC-TT-3′(forward primer) and 5′-GCG-CGG-AGT-ATC-AAT-AGA-GC-3′ (reverse primer) [28]. The MCMV plasmid was used for the development of the qPCR calibration curves and for the absolute quantification of the MCMV viral load, which was kindly provided by Dr. Julie Dechanet-Merville (CNRS UMR 5164, University of Bordeaux, France). The limit of detection in mouse blood was of 1.42 MCMV DNA copies (cp)/µL (95% CI: [1.26–1.61]), and the limit of quantification was 10 cp/µL of blood. 

On the day of euthanasia (D37), the small intestine and colon were dissected, homogenized and then filtered (100 µm filter) in a protective agent for nucleic acids (RNAprotect Cell Reagent, Qiagen, Hilden, Germany). DNA extraction was performed as previously described for human biopsies [16]. To verify tissue integrity and relate the viral load to a cell count, a fragment of the gene encoding the murine glyceraldehyde 3-phosphate dehydrogenase (GAPDH) was also amplified using the following primers: 5′-GCT-TGC-TGA-TGA-ATG-AGT-TC-3′ (forward primer) and 5′-CCT-GGG-AAG-TTT-GTT-CCA-3′ (reverse primer). M2-10B4 cells were used for cell count calibration. The tissue viral load was expressed in number of copies by 100,000 cells as used in humans diagnosis [29]. The limit of detection in mouse tissue was of 1 MCMV DNA cp/100,000 cells, and the limit of quantification was 5 cp/100,000 cells.

### 2.9. Statistics

Statistical analysis was performed with GraphPad Prism 5.03 software. A two-way ANOVA test with Bonferroni correction or a one-way ANOVA test with Bonferroni correction was used for the comparison of means between the groups of mice. *p* < 0.05 was considered statistically significant.

### 2.10. Ethical Considerations

Animal experiments were conducted according to the European Union rules on animal welfare. Prior to starting the experiments, the protocol was approved by the local ethical committee (CEEA-Loire) and the Animal Welfare Committee of the PLEXAN (agreement 2017011315316714).

## 3. Results

### 3.1. Monitoring of MCMV Systemic Infection

Systemic dissemination of MCMV was controlled by measuring the blood viral load over time. Blood samples were analyzed at D7, D14, D21, D30, D37 p.i. (Figure 1). As expected, groups 1 and 2, which were not infected, tested negative for blood MCMV DNA at D7 (data not shown). Groups 3 and 4 that were infected with MCMV exhibited blood MCMV DNA with a maximum viral load at D14 (2 × 10^4^ cp/µL) followed by a progressive resolution of the infection after D21, until the lowest viral loads were observed from D30 (150 cp/µL). No significant difference in viral load or kinetics was observed between groups 3 and 4 regardless of exposure to DSS at D30. (Figure 2).

An inflection of the weight curves was observed in mice infected with MCMV, particularly between D3 and D10, compared to uninfected ones (data not shown). Nevertheless, the four groups exhibited no significant difference in weight loss at D30, just prior to the administration of DSS. This is indicative of a good recovery in infected mice at this time, which was expected.

### 3.2. Clinical Assessment of DSS Administration 

After administration of DSS at D30 in groups 2 and 4, BALB/c mice were monitored daily by longitudinal assessment of weight loss, stool consistency and fecal blood. A statistically significant weight loss was recorded in mice exposed to DSS at D36 (group 2: *p* < 0.05) and D37 (group 2: *p* < 0.001; group 4: *p* < 0.05) by comparison to non-exposed groups. However, no statistically significant difference in weight loss was observed between groups 2 and 4 (Figure 3A).

At D34, i.e., 4 days after the first administration of DSS, the mean of DAI in mice exposed to DSS was significantly higher than in not exposed ones (*p* < 0.001). This difference was maintained until the end of the experiment with an increase in the mean DAI from 2.2 at D34 to 5.2 at D37 (*p* < 0.001) (Figure 3B). However, no difference of DAI was observed between the two groups exposed to DSS (Figure 3B), which suggests that a previous MCMV infection did not worsen the DSS-induced enteritis.

### 3.3. Severe Intestinal Inflammation Is Induced by DSS, Regardless of MCMV Infection Status

As shown in Figure 3C, a significant decrease in colon length was observed in mice treated by DSS (groups 2 and 4) by comparison to controls (group 1) (*p* < 0.01). The same pattern was observed between MCMV-infected groups exposed or not to DSS (groups 3 and 4) (*p* < 0.05). No significant difference in colon length was observed between groups 1 and 3, nor between groups 2 and 4. Together, these data suggest similar levels of intestinal inflammation regardless of MCMV infection. Therefore, shortened colon length was solely due to DSS exposure and not to MCMV infection. 

In addition, the administration of DSS induced a significant increase in fecal lipocalin concentration in mice exposed to DSS by comparison to non-exposed ones, both at D3 (*p* < 0.01) and D6 (*p* < 0.001) post-administration (Figure 3D). By contrast, no statistically significant difference was observed between groups 2 and 4 at both times.

The Nancy index of each mouse is represented in Figure 3E (small intestine) and Figure 3F (colon) at D37 (day of euthanasia). A Nancy index > 3 was observed in most DSS-treated animals (groups 2 and 4), which corresponds to severe mucosal damage with ulcerations. Interestingly, the mean Nancy index was not significantly different between the two DSS-exposed groups, despite the fact that mice harbored different MCMV statuses (Figure 3E,F). No mucosal inflammation was recorded in mice that were not exposed to DSS, including those of group 3 that were MCMV-infected (Figure 3E,F).

### 3.4. Absence of MCMV Reactivation in DSS Model of Gut Inflammation 

The results of previous experiments indicated that infected mice exhibited no difference with non-infected ones. Together, the data suggest that previous MCMV acute infection had no influence on the level of inflammation induced by DSS administration. Those findings could be explained either by the absence of MCMV reactivation after DSS administration or by the absence of the role of MCMV reactivation in the DSS model of gut inflammation. To address this question, the MCMV tissue viral load was quantified in the small intestine (Figure 4A) and colon (Figure 4B) at D37. 

A detectable viral load was recorded in samples from the majority of mice from groups 3 and 4. Of note, the tissue viral load was always below the quantification threshold. These results suggest the persistence of low levels of MCMV in digestive tissues after the resolution of acute infection. Additional experiments were conducted on digestive tissues for elucidating whether MCMV DNA that was detected by PCR was replicative, even at low level. To achieve this goal, we tested the presence by immunohistochemistry of the IE1 early protein of MCMV in small intestine and colon specimens collected on D37 from the mice of each group. In addition, to show negative and positive controls, Figure 4C illustrates that the small intestine and colon specimens of these animals were all negative for IE1 labeling, which suggests an absence of MCMV expression in digestive tissues on D37 p.i., and finally a failure of our model to reactivate MCMV in gut mucosa by DSS-induced enteritis.

## 4. Discussion

HCMV reactivation in the digestive tissues of UC patients is a significant cause of severe colitis [11,12,13,14]. This study aimed to develop a mouse model of MCMV reactivation following DSS-induced enteritis that could mimic UC flares and help to understand the putative role of tissue inflammation in viral reactivation. 

Mice are models used to explore the pathophysiology of CMV and the antiviral immune response. Their natural susceptibility to MCMV allows studying viral latency, establishment of reservoirs and reactivation [30]. After resolution of the acute infectious episode in BALB/c mice, the viral genome remains detectable in blood, lungs [31], spleen [32] and kidneys [33]. Then, latent MCMV can be reactivated with systemic dissemination of the virus, notably in an immunosuppressive context, such as SCID mice [34]. However, we opted for an immunocompetent wild-type model because it was closer to the situation of UC patients, at least at the early stage of disease. BALB/c mice were preferred over C57BL/6J mice because of their Th2 immunological orientation [35] as found in UC patients [36]. In addition, TCR-α-deficient mice, known to develop intestinal inflammation and chronic colitis at 4–6 months of age [37], develop colitis with conflicting results regarding MCMV reactivation. Indeed, the presence of viral proteins was detected in immunohistochemistry (M40 protein), while viral titration and viral load in the tissue were both negative [38]. We did not use this model because, as for SCID mice, it is not representative of UC patients due to the significant immunosuppression induced by TCR-α deficiency.

In the DSS-induced enteritis model, intestinal inflammation results from the damage of intestinal epithelium cells, allowing the entry of endoluminal bacteria into the mucosa [21]. DSS also induces histological changes, such as ulcerations and infiltration by granulocytes of lamina propria and submucosa, all being found in UC patients [21]. Moreover, markers of active expression (mRNA and protein) of tumor necrosis factor alpha (TNFα) were shown to increase in the intestinal tract of mice after DSS administration [39,40], which is of particular interest because of the major role of this cytokine in the MCMV and HCMV reactivation [41]. All these findings pleaded for the relevance of this model for modeling colitis flare-ups.

However, if the DSS administration induced a severe inflammation at the intestinal level with important clinical and biological consequences, no difference was seen between groups 2 (DSS only) and 4 (DSS + MCMV), as if the previous MCMV infection had no influence on the DSS-induced inflammation. In addition, the impact of the DSS exposition on the colon length and the fecal lipocalin level, in relation with TNFα synthesis [24], was comparable between the two groups, which supports solely the involvement of DSS in the induction and severity of enteritis without any detectable role for MCMV. The histological assessment by the Nancy index confirmed that MCMV did not aggravate the DSS-induced enteritis in group 4. Despite a persistent viral load in digestive tissues at the end of DSS exposition, we observed viral loads below the qPCR quantification threshold and the absence of IE1 expression, which evidences that the model did not favor MCMV reactivation during DSS-induced enteritis. Perhaps a lower dose of DSS and repeated cycle of DSS induction might have been used to decrease tissue damage caused by DSS exposition and to get closer to a chronic model of inflammation. 

Our results are in discordance with the work of Onyeagocha et al., who showed that MCMV infection, whether it be in an acute or latent state, exacerbated the severity of colitis induced by DSS [19]. They used C57BL/6 mice that are less sensitive to MCMV than BALB/c mice because of the expression of the Ly49H lectin receptor on NK cells [42]; consequently, they applied a dose of 10^5^ PFU of MCMV to generate viral infection by IP route [19]. Of note, this high MCMV dose killed the BALB/c mice very fast in our hands (data not shown); we determined the dose of 3 × 10^3^ PFU according to the criteria of lethality, animal welfare and good infectivity. A higher dose would perhaps have allowed greater intestinal dissemination but with an increased risk of mortality and animal suffering. These authors found also more severe colitis with weight loss, shortened colon length, intestinal crypt damage, and inflammatory cell infiltration. However, they did not find a significant difference in tissue viral load in the MCMV/DSS group compared to the MCMV group [19], which raises doubts about the impact of MCMV located in digestive tissues on the exacerbation of colitis. In another study, Brunson et al. infected C57BL/6J mice with 3 × 10^4^ PFU and added DSS diluted at 3% in the drinking water [20]. They showed an accelerated development of DSS-induced colitis with an early onset of gross bleeding in mice of the MCMV/DSS group compared to the DSS group. However, they did not monitor the viral load in gut specimens, which did not allow to assess MCMV reactivation in digestive tissues.

Models of digestive tract inflammation in wild-type mice are induced with other chemical agents, such as trinitrobenzene sulfonic acid (TNBS) or oxazolone [43]. TNBS-induced colitis mimics CD patients [44], which does not fit our UC-like model. Oxazolone appears to be an interesting chemical agent for modeling UC-like colitis, even if this colitis was shown to be essentially dependent on Interleukin-13-producing NKT cells [45], certainly involved in the pathophysiology of UC, but without being the only ones. Additionally, histone deacetylase inhibitors known to transiently induce viral lytic gene expression of HCMV [46] might be useful to induce MCMV reactivation [47], although it remains to be tested in the context of UC-like colitis. More globally, are wild-type mouse models infected with the Smith strain of MCMV the best ones for studying viral reactivation? The use of humanized mice, permissive to HCMV infection [48], or of non-human primates [49] would probably offer more appropriate models for studying the role of HCMV reactivation in the pathophysiology of UC flares.

## 5. Conclusions

Our results have shown that the model proposed in this study is characterized by a systemic MCMV infection allowing dissemination of the virus in the digestive tissues. However, DSS-induced acute enteritis 30 days after the primary MCMV infection failed to induce viral reactivation in the digestive tract. Although the inflammation was significant and the tissue damages were important, the model failed to document any role of MCMV in the occurrence of the gut lesions. More research is needed to set up pertinent animal models that could help to understand more clearly the involvement of HCMV reactivations in the course of UC and to evaluate therapeutic approaches that could improve the long-term evolution of this incurable disease.

## Figures and Tables

**Figure 1 viruses-14-02595-f001:**
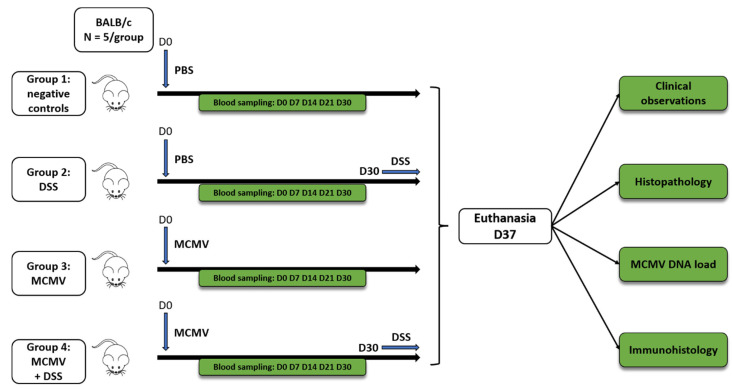
Flowchart of the study. Seven-week-old female BALB/c mice were divided into four groups (5 animals per group): group 1 = negative controls; group 2 = mice treated by dextran sodium sulfate (DSS) in the drinking water from D30 to D37; group 3 = mice infected by murine cytomegalovirus (MCMV) at D0; and group 4 = mice infected by MCMV at D0 then treated by DSS in the drinking water from D30 to D37.

**Figure 2 viruses-14-02595-f002:**
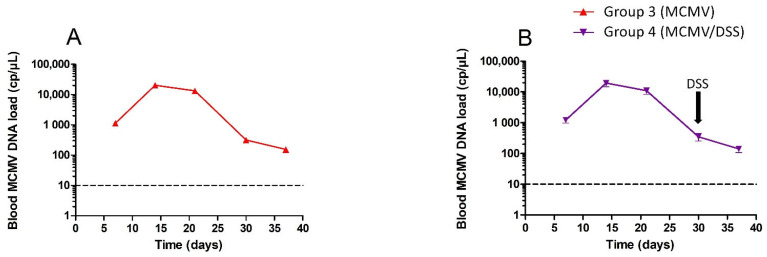
Kinetics of blood MCMV DNA load measured by qPCR for infected mice. Group 3 (red curve, (**A**)) and group 4 (purple curve, (**B**)). D0 corresponds to the time of infection with MCMV and D37 represents the time of sacrifice. The arrow denotes the beginning of dextran sodium sulfate (DSS) treatment in group 4. The limit of PCR quantification is represented by the horizontal line. The statistical test that was used was a two-way ANOVA with Bonferroni correction for panels. Standard variations are too low to be seen in the figures.

**Figure 3 viruses-14-02595-f003:**
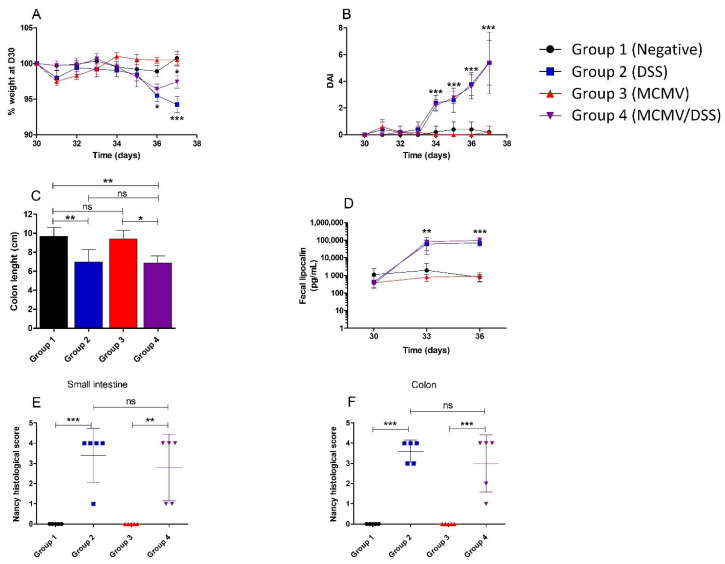
Monitoring of weight loss, clinical score and gut inflammation during DSS-induced colitis of MCMV-infected mice. Monitoring of clinical, biological and histopathological parameters. Mice were infected at D0 or not by murine cytomegalovirus (MCMV) and treated or not by dextran sodium sulfate (DSS) from D30 to D37 post-infection (p.i.) before being euthanized. (**A**)—Longitudinal weight loss assessment from D30 to D37. (**B**)—Evolution of the mean Disease Activity Index (DAI) from D30 to D37 p.i. (**C**)—Length (in cm) of the colon at the time of sacrifice, expressed as a mean. (**D**)—Longitudinal assessment of faecal lipocalin levels (in pg/mL) post treatment by DSS, values are expressed as means. (**E**)—Determination of the Nancy histological score (from 0 to 4) of the small intestine specimen from individual mice at sacrifice (D37 p.i.). (**F**)—Determination of the Nancy histological score (from 0 to 4) of the colon specimen. Statistical tests that were used were a two-way ANOVA with Bonferroni correction for panels (**A**,**B**,**D**), and a one-way ANOVA for panel (**C**,**E**,**F**). ns: not statistically significant, * *p* < 0.05, ** *p* < 0.01, *** *p* < 0.001.

**Figure 4 viruses-14-02595-f004:**
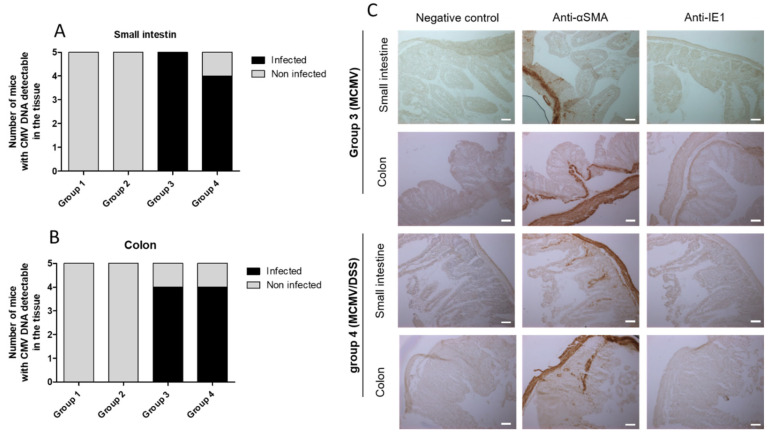
Exploration of MCMV reactivation in the gut of mice. (**A**)—Proportion of mice for which murine cytomegalovirus (MCMV) DNA was detected in small intestine specimens by PCR; positive mice are shown in black and negative ones in grey. (**B**)—For colon specimens. (**C**)—Immunohistochemistry at D37 (day of euthanasia) on gut specimens of a mouse of group 3 (upper part) or of group 4 (lower part). Staining corresponds to positive zones marked by horseradish peroxidase and revealed by diaminobenzidine substrate. The negative control was performed by omitting the primary antibody. The positive control was performed by using anti-alpha smooth muscle actin (anti-αSMA) as the primary antibody. The presence of a replicative virus was looked for by using an anti-immediate early (IE) 1 protein of MCMV as primary antibody. The scale bar on images corresponds to 50 µm. Photographs were taken at 20× objective in Zeiss Axioimager Apotome 3 optical microscope with a Zeiss quadriCCD Axiocam camera.

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
