# Peer review of "Impact of Dextran-Sodium-Sulfate-Induced Enteritis on Murine Cytomegalovirus Reactivation"

_viruses, 2022, doi:10.3390/v14122595_

Round 1
Reviewer 1 Report
Clinical study results have reported that CMV infection (reactivation) contributes to refractory ulcerative colitis. Results from mouse models support this association.
In this study, DSS was administered to MCMV-infected BALB/c mice, and the effect of its reactivation on intestinal inflammation was examined.
Acute intestinal inflammation induced by DSS administration did not reactivate MCMV in colonic tissue. As a result, the presence or absence of infection did not contribute to the exacerbation of intestinal inflammation.
As the authors know well, DSS acute colitis is unsuitable for a UC model. Therefore, the effect of this CMV reactivation on the pathogenesis of UC should be examined in a chronic colitis model.
Author Response
Clinical study results have reported that CMV infection (reactivation) contributes to refractory ulcerative colitis. Results from mouse models support this association.
In this study, DSS was administered to MCMV-infected BALB/c mice, and the effect of its reactivation on intestinal inflammation was examined.
Acute intestinal inflammation induced by DSS administration did not reactivate MCMV in colonic tissue. As a result, the presence or absence of infection did not contribute to the exacerbation of intestinal inflammation.
As the authors know well, DSS acute colitis is unsuitable for a UC model. Therefore, the effect of this CMV reactivation on the pathogenesis of UC should be examined in a chronic colitis model.
As rightly said, DSS acute colitis is unsuitable for a chronic colitis model.
As mentioned in the introduction, the gut inflammation was induced by the oral administration of dextran sodium sulfate (DSS) that acts as a trigger of acute enteritis.
Chronic colitis model such as TCR-α deficient mice induce significant immune deficiency which is not representative of the level of immunosuppression of patients with UC. We have modified several sentences in order to insist on the acute colitis: lines 63, and 331-333
Reviewer 2 Report
This manuscript focusses on an important topic, which will doubtless be of interest to readers. The attempt to extend useful animal models for cytomegalovirus disease is a worthy research goal and the decision to publish negative results is laudable.
The paper is well put together. There are some mistakes in the English that would benefit from the attention of a native speaker, but it is in general understandable. The figures are clearly made and clearly explained and show what is described.
There are only a few issues, one major, which I would like to see addressed in this otherwise very good publication.
Results:
Section 3.2 line 207. “the measure of the 0” appears to be a mistake.
Figure 3: The title of this figure is not informative.
Figure 4: This experiment is central to the conclusions of the paper and does not include a positive control. DSS induced reactivation is not seen by IHC for IE1, however no positive control for reactivation or even for IE1 expression is included. This seriously undermines the paper and should be addressed, particularly, as is mentioned in the discussion, as these results contradict those of other groups. I appreciate that repeating the entire experiment is probably not possible; would it however be feasible to reanalyse these gut samples in parallel with others where IE1 is expressed simply to show that the staining is working? Some effort to address this point should be made.
Conclusions: The final sentence has been left in the manuscript, presumably unintentionally.

Author Response
This manuscript focusses on an important topic, which will doubtless be of interest to readers. The attempt to extend useful animal models for cytomegalovirus disease is a worthy research goal and the decision to publish negative results is laudable.
Thank you very much
The paper is well put together. There are some mistakes in the English that would benefit from the attention of a native speaker, but it is in general understandable. The figures are clearly made and clearly explained and show what is described.
The english language has been reviewed especially by SFD, a canadian native researcher
There are only a few issues, one major, which I would like to see addressed in this otherwise very good publication.
Thank you very much
Results:
Section 3.2 line 207. “the measure of the 0” appears to be a mistake.
The sentence has been modified by “After administration of DSS at D30 in groups 2 and 4, BALB/c mice were monitored daily by measure the weight loss, stool consistency and faecal blood.” Line 210
Figure 3: The title of this figure is not informative.
The tittle has been changed by: “Monitoring of weight loss, clinical score and gut inflammation during the experiment”. Lines 216-217
Figure 4: This experiment is central to the conclusions of the paper and does not include a positive control. DSS induced reactivation is not seen by IHC for IE1, however no positive control for reactivation or even for IE1 expression is included. This seriously undermines the paper and should be addressed, particularly, as is mentioned in the discussion, as these results contradict those of other groups. I appreciate that repeating the entire experiment is probably not possible; would it however be feasible to reanalyse these gut samples in parallel with others where IE1 is expressed simply to show that the staining is working? Some effort to address this point should be made.
We totally agree with your comment. A positive control of IHC experimentation should be added. In order to prove that IE1 labelling can reveal CMV infection we propose to add a supplementary figure (in Methods: 2.7. Immunohistochemistry) with positive MCMV labelling of mouse salivary gland. We know that it is not the same tissue but MCMV administered by intraperitoneal (IP) is replicative at D21 in the salivary glands. It is therefore probably the best positive control for our model.
Conclusions: The final sentence has been left in the manuscript, presumably unintentionally.
Indeed, it was a mistake, thank you for your vigilance: the sentence has been deleted.
Round 2
Reviewer 1 Report
none
Reviewer 2 Report
Thank you very much for making the suggested changes.